# Colloidal quantum dot molecules manifesting quantum coupling at room temperature

Jiabin Cui [1,2,3], Yossef E. Panfil[1,2,3], Somnath Koley[1,2,3], Doaa Shamalia[1,2], Nir Waiskopf[1,2], Sergei Remennik[2], Inna Popov[2], Meirav Oded[1,2] & Uri Banin [1,2]*

Coupling of atoms is the basis of chemistry, yielding the beauty and richness of molecules. We utilize semiconductor nanocrystals as artificial atoms to form nanocrystal molecules that are structurally and electronically coupled. CdSe/CdS core/shell nanocrystals are linked to form dimers which are then fused via constrained oriented attachment. The possible nanocrystal facets in which such fusion takes place are analyzed with atomic resolution revealing the distribution of possible crystal fusion scenarios. Coherent coupling and wave-function hybridization are manifested by a redshift of the band gap, in agreement with quantum mechanical simulations. Single nanoparticle spectroscopy unravels the attributes of coupled nanocrystal dimers related to the unique combination of quantum mechanical tunneling and energy transfer mechanisms. This sets the stage for nanocrystal chemistry to yield a diverse selection of coupled nanocrystal molecules constructed from controlled core/shell nanocrystal building blocks. These are of direct relevance for numerous applications in displays, sensing, biological tagging and emerging quantum technologies.

[1] Institute of Chemistry, The Hebrew University of Jerusalem, Jerusalem 91904, Israel. [2] The Center for Nanoscience and Nanotechnology, The Hebrew University of Jerusalem, Jerusalem 91904, Israel. [3] These authors contributed equally: Jiabin Cui, Yossef E. Panfil, Somnath Koley. *email: uri.banin@mail.huji.ac.il

Colloidal semiconductor quantum dots (CQDs) that contain hundreds to thousands of atoms have reached an exquisite level of control, side by side with gaining fundamental understanding of their size, composition, and surface-controlled properties leading to their implementation in technological applications[1]. The strongly quantum confined energetic levels of CQDs possess atomic like character, for example—*s* and *p* states, related to their spherical symmetry. This, alongside with the ability to manipulate CQDs into more elaborate structures, naturally led to their consideration as "artificial atoms". Inspired by molecular chemistry, in which functionality of molecules depends on how atoms couple, we apply analogous concepts to enrich CQD-based materials. If one considers CQDs as artificial atom building blocks[2,3], how plentiful would be the selection of composition, properties, and functionalities of the corresponding artificial molecules? Herein we introduce the utilization of CQDs as basic elements in "nanocrystal chemistry" for construction of coupled colloidal nanocrystals molecules focusing on homodimer quantum dots (QDs), in analogy to homonuclear diatomic molecules.

Coupled QDs were prepared by means of molecular beam epitaxy (MBE)[4–6]. However, MBE-grown double QD structures exhibit some limitations. First, the size of MBE-grown QDs is larger than the colloidal ones, and the typically large distance between the QDs limits wave-function tunneling that yields coupling phenomenon. Correspondingly, such structures exhibit wave-function tunneling that typically yields coupling energies of a few meV confining their utility to low-temperature operation in specialized cryogenic applications[7,8]. Furthermore, MBE-grown structures are inherently buried within a host semiconductor[9]. In contrast, colloidal QDs are free in solution and accessible for wet-chemical manipulations through their surface functionalization. Using such knobs, CQD molecules were constructed by connection with DNA strands providing geometrical control[10], yet in such structures the linker DNA molecules form a barrier that minimizes quantum mechanical coupling. Addressing this limitation, core/multishell structures with concentric regions were first examples of coupling within CQDs architectures, where the wave functions of two well regions within such NCs may interact leading to CQDs showing dual emission peaks[11]. Other examples constitute synthesis of dot-in-rod structures and growing an additional QD region on the rod apex, thus yielding a coupled system[12] and dumbbell architectures[13]. However, these progresses were either restricted by specific morphologies[14], specific materials, and relatively large coupling barrier distance and height[15–17]. Therefore, there is a lack of a general approach for producing coupled CQD molecules in which there is flexibility to tailor the potential energy landscape and to tune the coupling strength.

Herein, we introduce a facile and powerful strategy for coupled CQD molecules with precise control over the composition and size of the barrier in between them to allow for tuning their electronic coupling characteristics and optical properties. This entails the use of core/shell CQDs as artificial atom building blocks. In terms of the band gap engineering, in first instance, tuning the core size is used to manipulate the wave functions and energies of the electron and hole. On top of this, further control has been afforded by the synthesis of shells on these cores. While the chemical bond is the basis for combining atoms in molecules, connecting CQDs has to occur through adjoining of their crystal faces to form a continuous crystal. Thus, fusing two core/shell CQDs yields a homodimer with a tailored barrier dictated by the shell composition, thickness, and fusion reaction conditions. With such control, using high-resolution aberration-corrected scanning transmission electron microscopy, we investigate the orientation relationships including homo-plane-attachment and

hetero-plane-attachment in the fusion process. Moreover, the manifestations of quantum coupling are revealed by the broadening and redshift of the band gap transitions along with modification in higher energy absorption bands, in agreement with the quantum mechanical calculations for the system. The emerging attributes of coupling are also revealed by single nanoparticle spectroscopy studies yielding modified electron–hole recombination rates and single photon statistics in CQD dimers in comparison to monomers. The approach introduced herein serves as a basis for a wide selection of CQD molecules utilizing the rich collection of the artificial atom core/shell CQD building blocks. Such CQD molecules bear significant promise for their utilization in numerous applications, including in light-emitting devices, displays, photovoltaics, and sensors. CQD molecules offer enhanced coupling efficiency by their smaller sizes and by the small distances between the QDs resulting in quantum mechanical coupling an order of magnitude larger than prior MBE-grown QD systems that is well resolved even at room temperature.

## Results

**Formation of coupled CQD molecules**. Exemplary coupled homodimer molecules were generated from CdSe/CdS core/shell[18,19] CQDs via a procedure utilizing silica nanoparticles (NPs) as a template for forming molecularly linked dimers[20], which are then fused via a high-temperature reaction (Fig. 1a, full scheme in Supplementary Fig. 1). Three different CdSe/CdS core/shell CQDs were studied (1.9/4.0, 1.4/2.1, and 1.2/2.1 nm core radius/shell thickness, see Methods and Supplementary Discussion for synthesis details and Supplementary Fig. 2 for transmission electron microscopy (TEM) images and optical spectra). The TEM and high angle annular dark field (HAADF) scanning transmission electron microscopy (STEM) characterization manifests the wurtzite structure of the monomer CdSe/CdS QDs (Supplementary Fig. 3). These CdSe/CdS CQDs were bonded via thiol linking to the surface of an SiO₂ NP template substrate (Supplementary Figs. 4 and 5). A second SiO₂ layer was grown for masking the remaining SiO₂ surface and to immobilize the bonded CQDs (Supplementary Fig. 6), followed by treatment with a tetrathiol linker (Supplementary Fig. 7). Adding a second CQD leads to formation of a molecularly linked dimer structure (Fig. 1b, Supplementary Fig. 8). Next, the SiO₂ template NPs were selectively etched by HF treatment. Size-selective precipitation was used to separate out the monomers and obtain a highly dimer-enriched sample (Fig. 1c).

The dimerization procedure yields a dimer structure with an organic insulating barrier. Hence, to achieve a coupled system, a last step of fusion is required. The fusion procedure was performed while adding Cd-oleate and heating to 180 °C for 20 h. Figure 1d, e presents the fused dimer structure after a size selection procedure (Supplementary Figs. 9 and 10). At this nontrivial important stage, the reaction parameters, including temperature, time and ligands type and concentration, have a significant influence on the coupled dimers formation. If the temperature was too high (above 240 °C), collapse of the dimer structures through linker bond cleavage may occur, as well as CQD ripening distorting the core/shell architectures. On the other hand, if the temperature was too low, the fusion rate would be too slow and inefficient. The dimer structure formation is also very sensitive to excess of ligands in the solution, which inhibits the fusion and leads to a decrease in the dimer yield. Therefore, careful tuning and choice of these reaction parameters is crucial for achieving high dimer yields and lower yields of dimer collapse and ripening, while achieving a continuous linking region of the shell materials forming the barrier between the two cores in

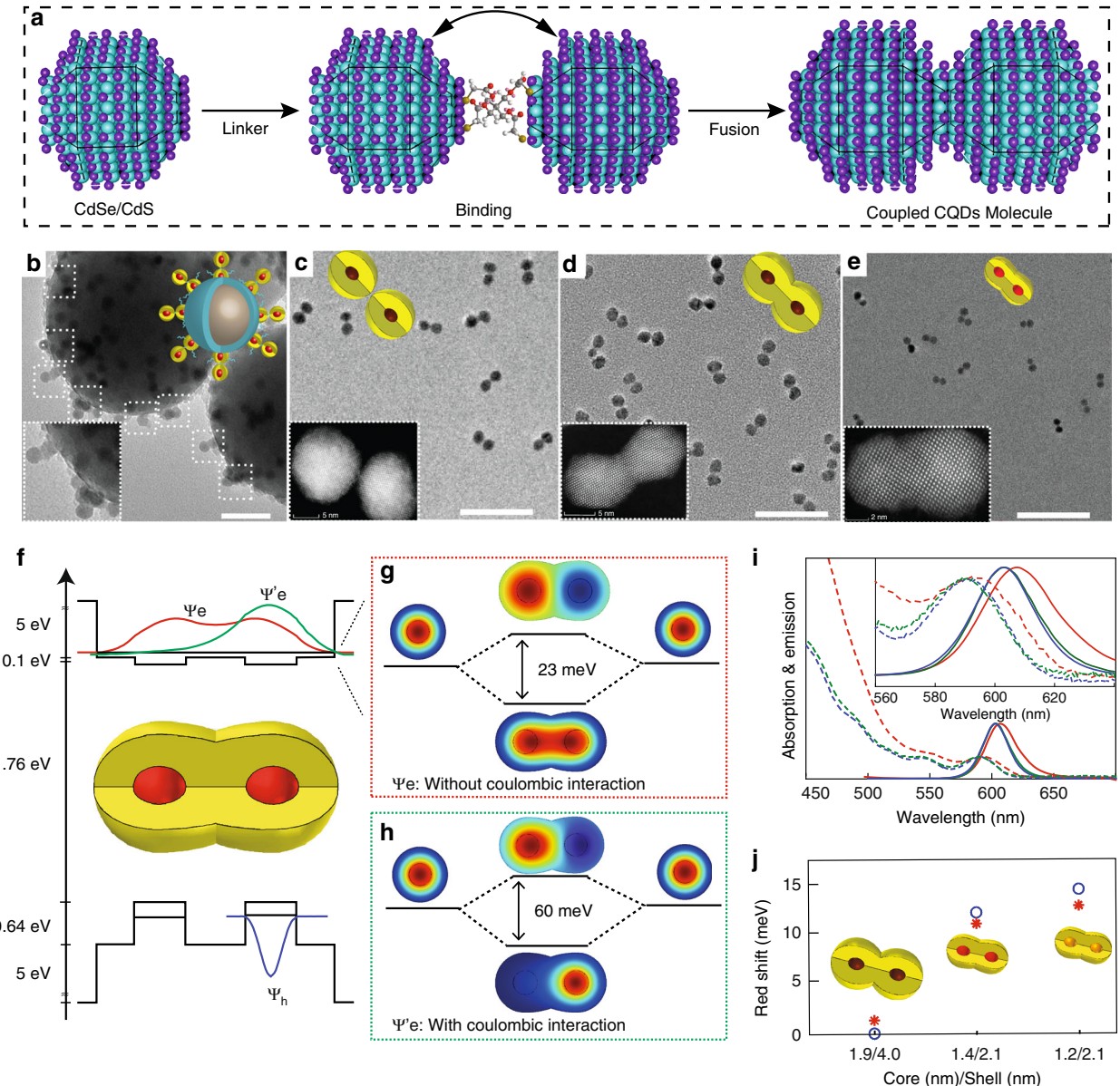

**Fig. 1** Coupled CQD molecules. **a** Scheme for fabrication of coupled CdSe/CdS CQD molecule. **b** The dimer@SiO₂ CQD structure. The dimer 1.9/4.0 nm CQD molecules **c** before and **d** after the fusion procedure. **e** The 1.4/2.1 nm fused CdSe/CdS CQD molecules. Schematic structures are illustrated. Scale bars (**b–e**) are 50 nm and insets show higher magnification images. **f** The potential energy landscape and a cross-section of the calculated first electron wave function without Coulombic interaction $\Psi_e$ (red), with Coulombic interaction $\Psi'_e$ (green) and of the hole wave functions $\Psi_h$ (blue) of the coupled CQD molecules. **g** Calculated bonding and anti-bonding two-dimensional electron wave functions without (cross-section of the bonding state is the red curve in **f**), and **h** with Coulombic interaction (cross-section of the bonding state is the green curve in **f**). **i** Absorption (dashed lines) and fluorescence spectra (solid lines) of monomers (blue), unfused (green), and fused 1.4/2.1 nm CdSe/CdS CQD molecules (red). **j** Calculated (red asterisk) and experimental (blue circles) band gap redshift of monomer-to-respective-homodimer structures for CQD molecules with different core/shell dimensions.

the fused dimers. Further considerations of this important fusion step and the resultant interfacial structures are discussed later.

**Optical signatures for coupling and wave-function hybridization.** After the fusion step, the resultant CQD dimer leaves an interesting optical signature of a redshift in the absorption and photoluminescence spectra along with broadening of the band gap and excited state spectral features (Fig. 1, Supplementary Fig. 11). Generally, there are several factors which can lead to a redshift: the formation of alloying shell[21], alteration of the dielectric environment[22] (surface ligands), or interfacial strain[23]. To address these different possibilities, we also studied the spectral properties of the

monomers, which underwent the fusion reaction under similar conditions (Supplementary Figs. 9 and 10), and found them to be identical to the original monomer particles (Supplementary Fig. 12). Hence the possibility of observing a redshift in the band gap transition due to formation of an alloy shell or altered dielectric environment can be ruled out. Furthermore, strain effects and change in the dielectric properties during the fusion procedure can be considered negligible as we did not grow an additional shell, but rather the fused shell material is the same (CdS), and the surface ligands are also the same for the CQDs monomers and dimers. Moreover, no shift was observed after the fusion of the large 1.9/4.0 nm CQDs (Fig. 1), where dielectric and strain effects, if significant, would be expected to contribute as

well. In fact, the redshift in the band gap transitions was found to depend systematically on the alteration of the core size and shell thickness of the monomer counterparts, increasing for small core and shell dimensions (Fig. 1j). This is consistent with the difference in the delocalization of the wave functions in the various CQDs that lead to different degree of coupling of the corresponding wave functions in the CQD molecules.

To this end, we have employed quantum mechanical calculations to visualize the wave-function hybridization and to calculate the expected redshift in the different CQD dimers. The changed potential energy landscape upon fusion leads to hybridization of the monomer QD wave functions in the dimers (Fig. 1f), in analogy to homonuclear diatomic molecules. We utilized finite element software (COMSOL) to calculate the energy levels and wave functions of the fused CdSe/CdS dimer and monomers within an effective-mass-based approximation (Supplementary Discussion and Supplementary Table 1 for details). The conduction band in this system is demonstrating the fundamental textbook example of hybridization. According to this model, when the distance between two atoms is decreased, their wave functions will hybridize to form a symmetric bonding state and anti-symmetric anti-bonding state with energy difference of twice the hopping energy. The bonding and anti-bonding electron wave functions, which, respectively, are in-phase and anti-phase superpositions of the monomer wave functions, are presented in Fig. 1g for the case of 1.4 nm core radius and a potential energy barrier between the dots of 4.2 nm (0.1 eV band offset), corresponding to the CQD molecules formed from 1.4/2.1 nm core/shell CQDs. Because of the quasi-type II nature of the CdSe/CdS interface, the monomers electron wave functions are easily hybridized and leading to 23 meV energy spacing between the bonding and anti-bonding electron states. For the hole, however, the valence band potential manifests a relatively high band offset of 0.64 eV, and this, combined with the heavier hole effective mass, yields minimal hole hybridization.

Considering the case of one exciton residing in the dimer and taking into account the Coulombic interaction between the electron–hole pair, since the hole wave function is essentially not hybridized, the hole is in one of the dots and consequently the electron does not see a symmetric double QDs' potential anymore. The calculated two lowest energy levels wave functions of the electron including the Coulombic interaction are presented in Fig. 1h. The Coulombic interaction for the first electron level, localized around the hole, is greater than the second electronic state in the opposite dot, increasing the energy spacing between the bonding and anti-bonding states to 60 meV. One can see that the electron is localized in the dot which contains the hole as well. However, there is still significant tunneling-coupling observed for the electron wave function and a redshift is predicted. This is indeed confirmed experimentally in the emission and absorption spectrum, where only in the case of the fused dimer a redshift is observed compared to the monomer (603–607 nm in case of 1.4/2.1 nm core/shell QD), whereas for the unfused organically linked dimer no redshift is seen (Fig. 1i, Supplementary Table 2). The control in the magnitude of the redshift, for the monomer to fused dimer transition, for three different types of CQDs is depicted in Fig. 1j. In the the case of 1.2/2.1 nm core/shell CQDs the redshift is increased (13/14 meV calculated/experimental) due to the greater spill out of the electron wave function to the shell because of the smaller core size. This is in contrast to the case of 1.9/4 nm core/shell CQD where the redshift is negligible (0.5/0 meV calculated/experimental) because of the localization of the electron wave function in the larger core (Supplementary Fig. 13, Supplementary Table 2).

An additional signature for the coupling in fused dimers is observed in the absorption spectra at higher energies. Figure 1i shows broadening only upon fusion consistent with coupling forming multiple states in dimers. Furthermore, the spectra normalized at the band gap manifest a significantly stronger relative absorbance in high energies for the fused dimers compared with monomers and unfused dimers (see also Supplementary Fig. 11). This is assigned to the wave functions modification in the fused system, which can be considered from a viewpoint of hybridization among the excited states.

**Structural characterization of fused CQD dimers**. We next consider further the non-trivial fusion stage and its consequences. Analysis by HAADF-STEM confirms that coupled dimer formation is indeed achieved based on fusion of the 1.9/4.0 nm core/shell QD monomers (Fig. 2). A continuous atomic lattice through the entire structure was formed upon fusing the two QDs shells (Fig. 2a). The core architecture in the coupled structure was maintained as demonstrated by the energy dispersive X-Ray spectroscopy (EDS) line scan measurement (Fig. 2b, c). A continuous distribution of cadmium (both in core and shell) and sulfur (only in shell) is identified along the line of the dimer axis. Along the same line, selective regions of the selenium (only in core) are clearly identified signifying the cores locations.

Underlying our fusion reaction strategy is the process of oriented attachment—a crystal growth mechanism in which secondary mono-crystalline particles can be achieved through oriented and irreversible attachments of primary particles[24–30]. PbSe CQD dimers were prepared via oriented attachment in solution, but even under succinct control of ligands, concentration, and reaction conditions, only ~30% dimer fraction, a rod-like colloidal quantum system, could be achieved along with monomers and higher order oligomers. In our template-based strategy, high control over dimer formation was achieved by firstly forming a connection by molecular linkers. The molecular linkers however constrain the initial relative crystal orientations between the two monomers. With careful tuning of precursor and judicious choice of the fusion condition, we can foresee high potential for this method to serve as a general coupling strategy for other colloidal nanocrystal systems. Here, we studied this special case of "constrained oriented attachment", and Fig. 3 shows exemplary orientation relationships observed for coupled molecules formed from the 1.9/4.0 nm CQDs and their detailed analysis. Both homo-plane and hetero-plane (misorientation) attachment relationships are observed. Homo-plane attachment orientation occurs via contact between homonymous faces (10$\bar{1}$0) and (10$\bar{1}$0), (0002) and (0002), (10$\bar{1}$1) and (10$\bar{1}$1) (Fig. 3a–c, g, i, respectively), consistent with the CQD monomer crystal model built based on STEM analysis (Supplementary Fig. 3). In such homo-plane attachment cases, both CQD monomers of a fused pair are projected under the same zone axis. This allows accurate identification of the fused faces at dimers orientated with its fusion axes normal to the projection zone axis (depicted in Fig. 3). Hetero-plane attachment orientation is observed at fusion of heteronymous faces: (0002)||(10$\bar{1}$0) (Fig. 3e), (0002)||(10$\bar{1}$1), (10$\bar{1}$1)||(10$\bar{1}$0) (Supplementary Fig. 14a, c).

The statistics of the orientation relationship within the CQD molecules is depicted in Fig. 3k (homonymous and heteronymous orientations are approximately equally abundant). The (10$\bar{1}$0)|| (10$\bar{1}$0) and (0002)||(0002) face attachments are dominant, whereas the (10$\bar{1}$1)||(10$\bar{1}$1) attachment is much less common. This is consistent with an interplay between the relative reactivity, surface passivation, and occurrence of the various faces on the monomer QDs. The (0002) facets, while in minority, manifest a Cd-rich termination with three dangling bonds per atom that can easily react with thiol linkers[31]. Both (10$\bar{1}$0) and (10$\bar{1}$1) facets are plentiful but better passivated[32,33]. However, linking to the (10$\bar{1}$1)

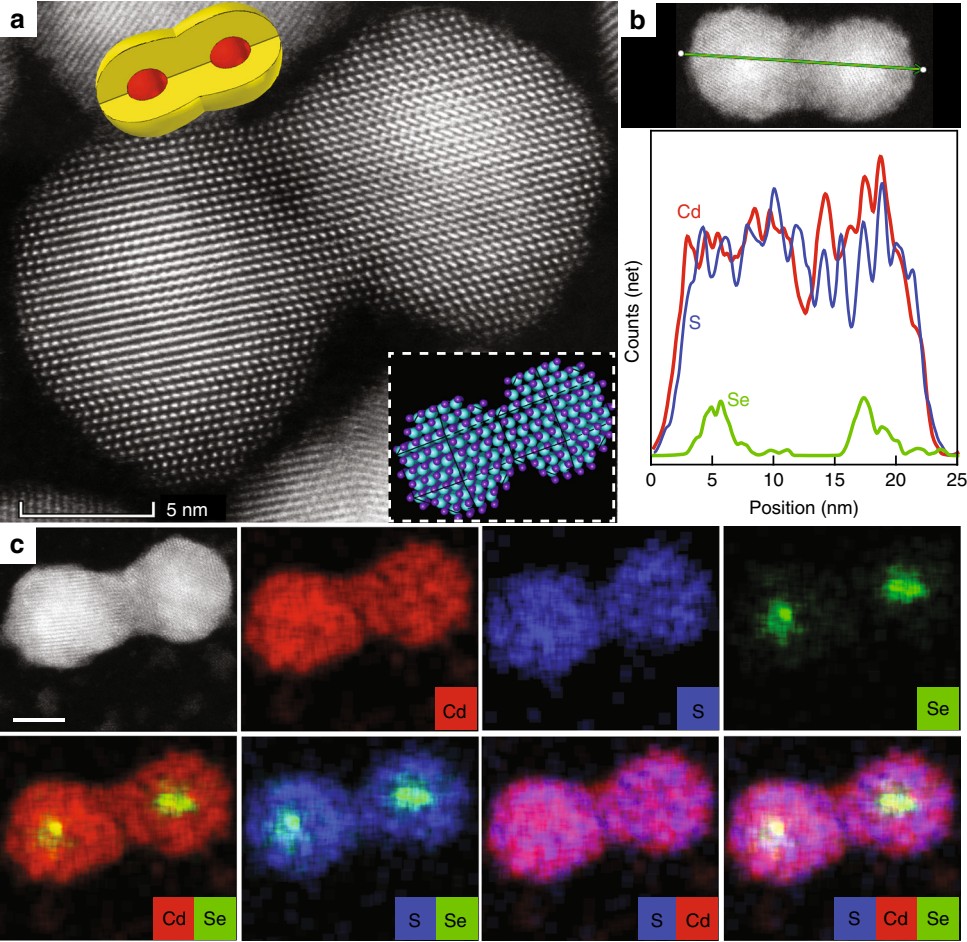

**Fig. 2** Dimer structure characterization and analysis. HAADF-STEM image (**a**), EDS line scan data (**b**), and STEM-EDS (**c**) analysis of the coupled 1.9/4.0 nm CdSe/CdS molecules. Inset presents an atomic model. Scale bars 5 nm.

facet is sterically hindered. The hetero-plane attachment statistics is also consistent with these considerations.

The generality of our formation strategy is well manifested also for the other CQD homodimers. Analytical and structural STEM analysis for the 1.4/2.1 nm core/shell CQDs are shown in Fig. 4 and Supplementary Figs. 15 and 16. Fusion conditions need to be tuned as smaller core/shell CQDs are more reactive. Therefore, shorter etching times were used, and also a shorter fusion process (~10 h).

**Photophysical characteristics of CQDs molecule.** Further manifestations of coupling are observed in the fluorescence properties studied both in ensemble and as single particles, comparing monomers with unfused and fused dimers. For the 1.4/2.1 nm CQDs the fluorescence decays vary, with monomers showing a nearly single exponential lifetime of 25 ns whereas fused dimers show nearly bi-exponential decay with a fast component of 5 ns (Supplementary Table 3). Unfused dimers show an intermediate behavior. Molecules of the larger 1.9/4.0 nm CQDs exhibit significantly smaller changes in lifetimes upon fusion, supporting the role of coupling in the variations observed for the 1.4/2.1 nm CQDs (Supplementary Fig. 17).

More detailed information is garnered from single NP fluorescence measurements on the 1.4/2.1 nm CQDs ("Methods" section for experimental details). The fluorescence from single monomers (Fig. 5b) exhibited a typical on–off bimodal distribution with a monoexponential fluorescence decay of the on-state[34] and strong photon antibunching[35]. In comparison, individual

fused dimers (Fig. 5c) under similar excitation conditions show more intense fluorescence consistent with their two-fold larger absorption cross-section. However, flickering of the fluorescence with a distribution of intensities is observed rather than distinct on–off states, accompanied by a significantly lower antibunching contrast (~0.75). The lifetime measurements indicate systematic shortening of the average single particle lifetimes from the monomers through the unfused to the fused dimers, in line with the ensemble measurements (Supplementary Fig. 18 for representative traces and statistics of the lifetimes). Analyzing the lifetimes of the high-intensity occurrences (green shaded regions in Fig. 5b, c) yields a significantly shortened average lifetime of 5 ns for the single fused dimer compared with the monomer (29 ns). It is noteworthy that within the fused dimers sample, we have detected ~15% of particles that have similar fluorescence characteristics as the CQD monomer sample, in line with their fraction from TEM analysis (Supplementary Figs. 19 and 20). This establishes that the fusion procedure in its entirety did not change the core/shell CQDs.

We consider two possible mechanisms related to coupling, both leading to shortening the lifetime in dimers. First, resonance energy transfer between the two dots (Fig. 5d), a mechanism nearly equally active for unfused and fused dimers. Second, tunneling of the electron to the other dot (Fig. 5d) as already illustrated in Fig. 1. Tunneling in unfused dimers occurs by collisional electron transfer and is strongly dependent on the linker[36], while in the fused dimer the potential barrier for tunneling is modified and reduced substantially. Both

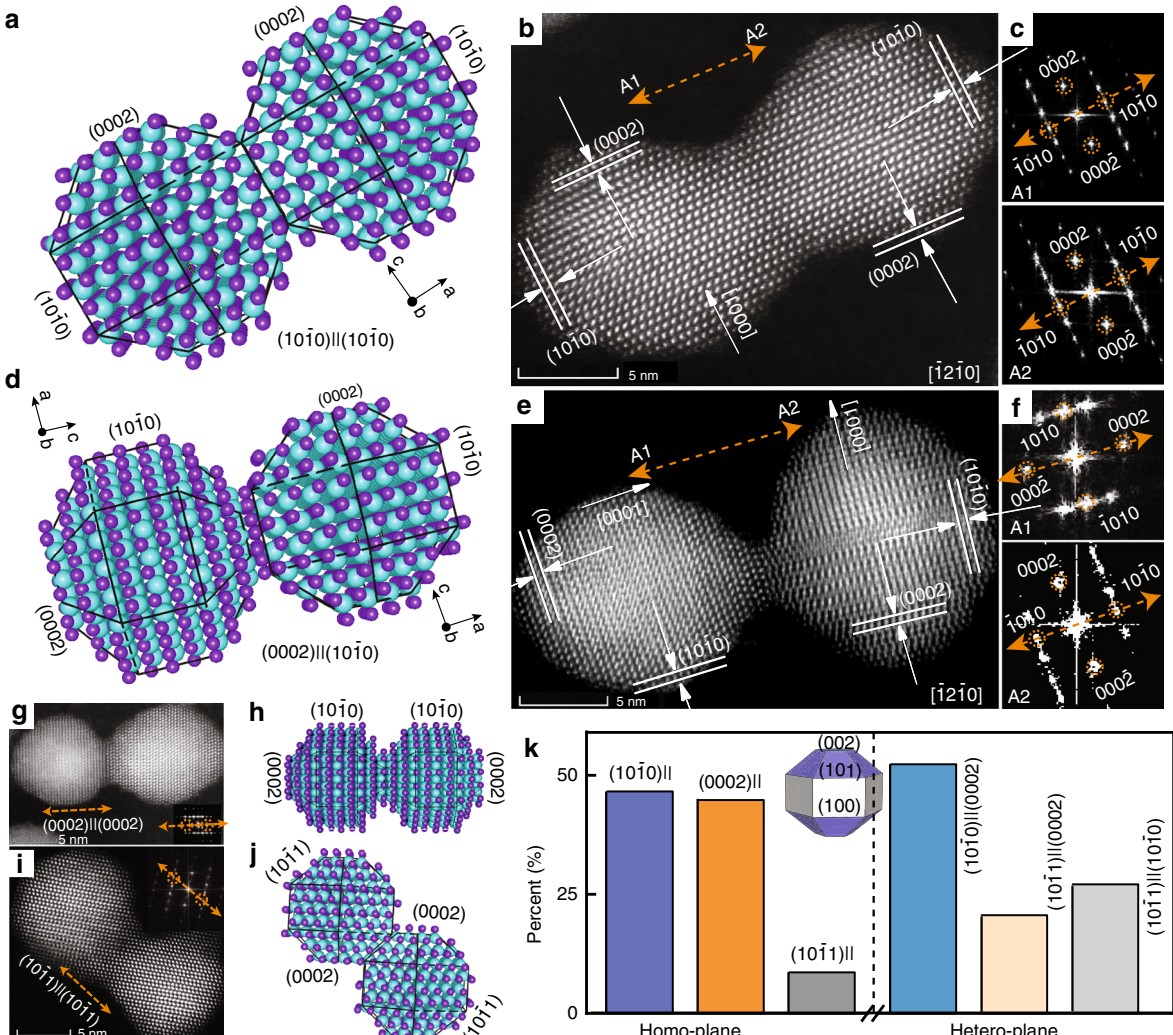

**Fig. 3** Fusion orientation relationships in CQD molecules. Atomic structure model (**a**, **d** cadmium atoms—brown, sulfur atoms—blue.), HAADF-STEM images (**b**, **e**), and FFT patterns (**c**, **f**) of the homo-plane (**a–c**) and hetero-plane (**d–f**) attachment of coupled CQD molecules with orientation relationship of attachment on (10$\bar{1}$0)||(10$\bar{1}$0) and (0002)||(10$\bar{1}$1), respectively. The HAADF-STEM (**g**, **i**) and atomic structure model (**h**, **j**) of homo-plane attachment on (0002) and (10$\bar{1}$1) facets, respectively. Dashed orange arrows indicate the CQD fusion/molecular axis in plane of the image normal to projection ZA [$\bar{1}$2$\bar{1}$0]. Note that for (10$\bar{1}$0)||(10$\bar{1}$0) homo-plane attachment, the homonymous (10$\bar{1}$0) faces of A1 and A2 are parallel (**c**), while for the hetero-plane attachment the heteronymous faces are parallel (0002)||(10$\bar{1}$0) (**e**). **k** Distribution of observed homo- and hetero-plane attachment orientations on (10$\bar{1}$0), (0002), and (10$\bar{1}$1) faces. Here, the total amount of dimers for the statistic was 100. Inset shows the CQD model and faces.

mechanisms will be enhanced for the smaller CQD dimers, but tunneling is more strongly dependent on the size/distances. Indeed, the large CQD molecules, where tunneling probability is negligible according to our calculations (Fig. 1), show smaller changes in lifetimes with shortening in the case of dimers compared to monomers and little differences before and after fusion (Fig. 5c, Supplementary Figs. 17 and 21) the notable lifetime shortening that is seen for the small CQD molecules upon fusion is indicative of the enhanced contribution of the tunneling mechanism in this case.

Next, we consider the photon statistics, which in CQDs are strongly influenced by multicarrier effects. An increment in the $g^2$(0) value was found in the case of dimers in general with the possibilities of either two emission centers in the excitation spot or intrinsic properties of coupled systems. Specifically, in the fused dimers, the particles absorb the light as one unit, and at low excitation regime ($<N> \sim 0.1$) the possibility of emission from one of the centers is rational statistically. The full understanding of the $g^2$(0) value requires more rigorous experimental studies while

an interesting multicarrier configuration can be realized in these fused CQD dimers. In dimer CQD molecules, a new type of biexcitons can occur, with each exciton occupying a different core (Fig. 5e). The large increase in the value of $[g^2 = \frac{\text{area}_{(0\,\text{ns})}}{\text{area}_{(200\,\text{ns})}} = \frac{\text{QY}_{(\text{BX})}}{\text{QY}_{(\text{X})}}$ at low excitation power] observed for the dimers versus monomers can be explained by this new type of biexciton for which the non-radiative Auger decay will be strongly suppressed increasing the biexciton quantum yield (QY) (Supplementary Fig. 22). Moreover, the single particle exciton QY will decrease for dimers on account of the tunneling of the electrons reducing the electron–hole overlap.

An additional difference relates to the fluorescence flickering in the dimers rather than distinct on–off fluorescence of the monomers (Fig. 5c, Supplementary Fig. 23). This indicates presence of multitude emitting configurations for dimers. Indeed, the lifetime traces for the high/low-intensity regions in the dimer are not single exponential. The low-intensity region is above background and not off and the lifetime has a ~5 ns component.

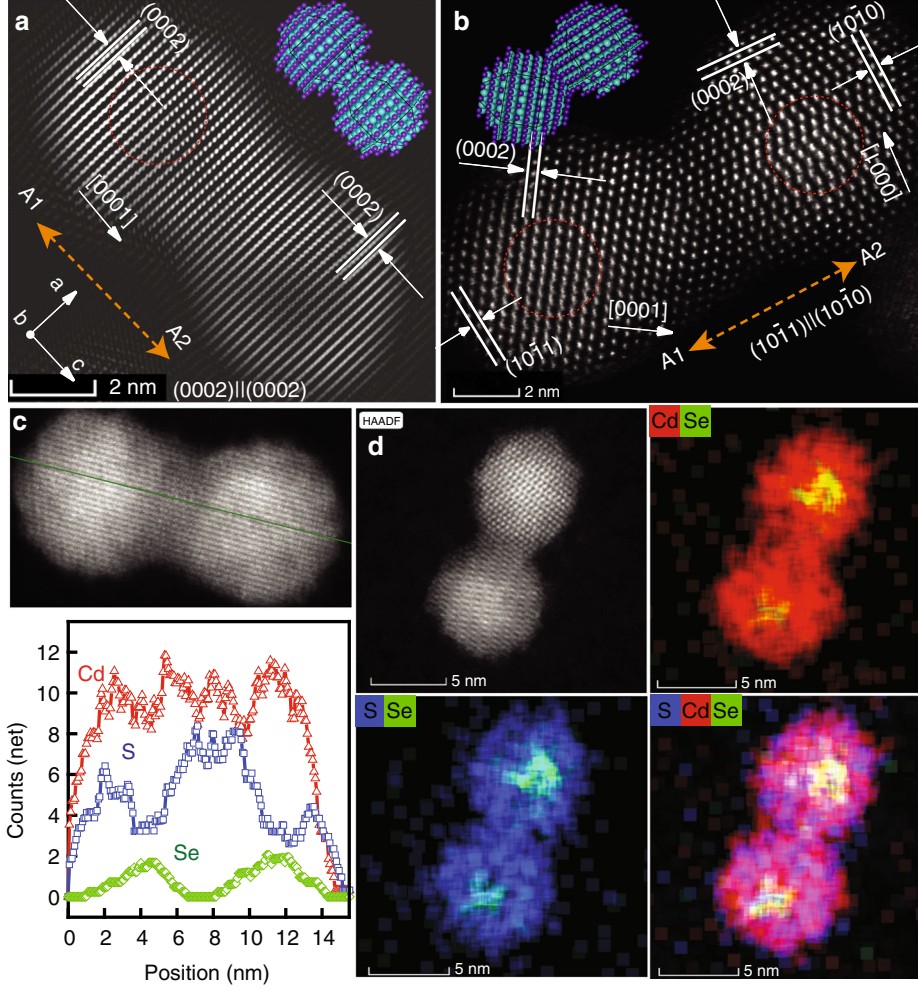

**Fig. 4** Characterization of 1.4/2.1 nm CQDs molecule. Fourier-filtered HAADF-STEM images of the coupled CdSe/CdS molecules with homo-plane attachment of (0002)||(0002) (**a**) and hetero-plane attachment of (10$\bar{1}$0)||(10$\bar{1}$1) faces (**b**). **c** EDS line scan data and **d** STEM-EDS analysis.

All this indicates to trion formation (positive or negative). In small CQD monomers, the trion states are strongly quenched by the Auger decay yielding an off state behavior. In dimers, which have large volume and the excess carrier may occupy the second dot region forming a new type of trion (Fig. 5e), the Auger rate is suppressed, and the trion can become emissive. Such an effect was reported for large CdSe/CdS core/shell manifesting gray state emission and is also observed by us for the large CQDs[37]. The multitude possibilities for emissive trion formation can explain the larger distribution of observed fluorescence intensities and the lifetime behavior for the dimers. To further address this point, an excitation intensity dependence was performed varying the average exciton occupancy <N> from 0.03 to 0.18 (Fig. 5f, Supplementary Fig. 24). The lifetime decreases upon increasing the laser power, indicating the increasing contribution of trion formation consistent with this description[34,38].

This study introduces a robust route to the realization of coupled colloidal QD molecules with high purity. The construction strategy for the CQD molecules utilizes well-controlled CdSe/CdS core/shell nanocrystal artificial atom building blocks constructing homodimers fused via "constrained attachment" of the nanocrystal facets. The synthesized novel CQD dimer structures, emitting in the visible range, exhibit electronic coupling at room temperature signified by the redshift and broadening of the band-edge transition observed both in absorption and in photoluminescence. This is attributed to the quantum coupling and hybridization of the monomer wave functions within the CQD molecules, as supported by quantum mechanical calculations. Broadening and modification of the absorption transitions at higher energies are also related to the coupling effects within the CQD dimer molecules. Furthermore, the photophysical properties of the CQD molecules at single NP level exhibit lifetime shortening, fluorescence flickering, and an increase in the g2 value for the photon antibunching compared to monomers. These characteristics indicate the introduction of additional recombination pathways and the rich possibilities for multiexciton configurations in the artificial CQD molecule compared with the monomers, also related to coupling within the system.

Formation of coupled CQD homodimer molecules sets the stage for "nanocrystal chemistry". Considering the rich selection of size and composition controlled CQDs emphasizes the analogy of these artificial atoms to atoms of the periodic table discovered by Mendeleev 150 years ago. As a future outcome, we foresee the formation of a diverse variety of coupled CQD molecules with prodigious promise for their utilization in numerous optoelectronic, sensing, and quantum technologies applications.

## Methods

**Reagents**. Oleylamine (OAm, 70%), oleic acid (OA, 99%), 1-octadecene (ODE, 90%), cadmium oxide (CdO, ≥99.99%), selenium (Se, 99.99%), trioctylphosphine

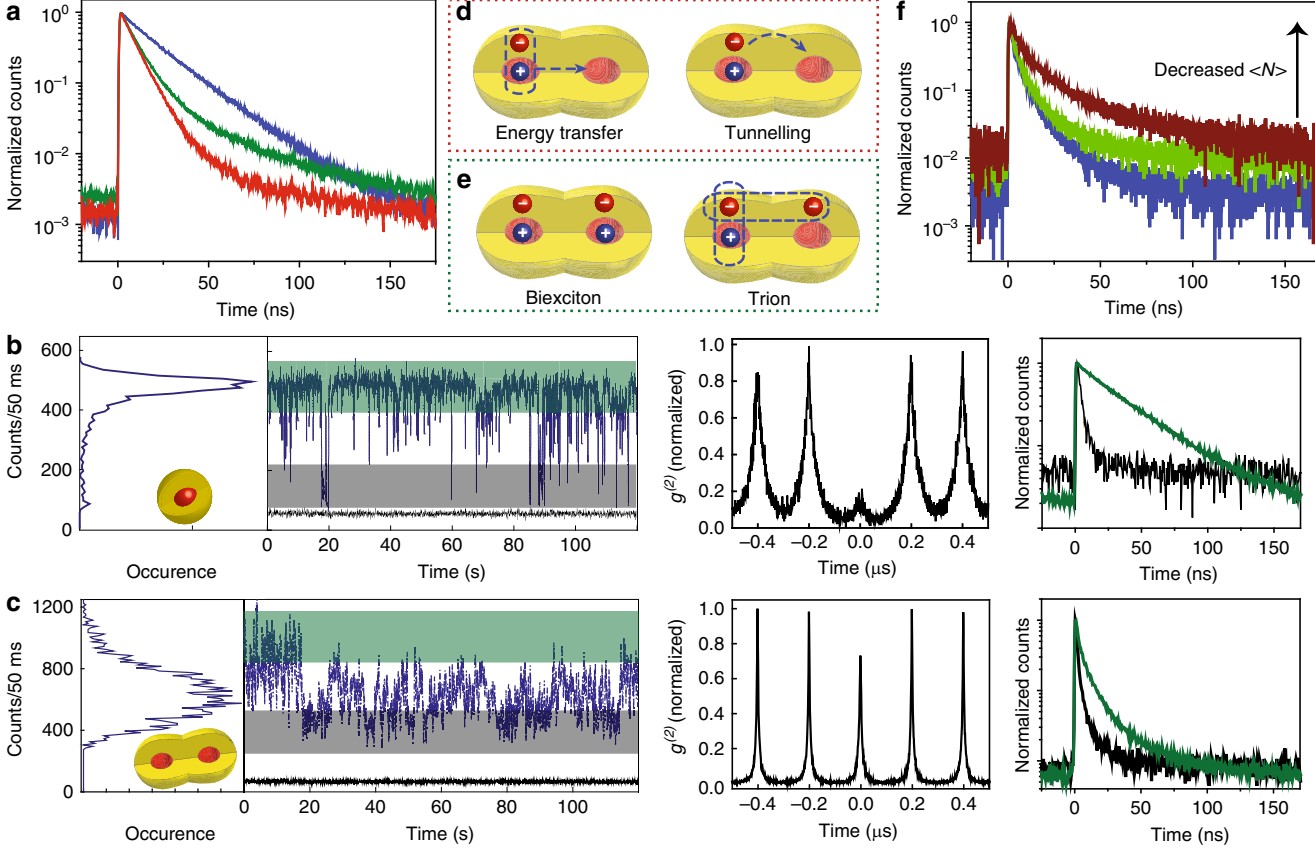

**Fig. 5** Coupling effects through fluorescence measurements for 1.4/2.1 nm CQDs molecules. **a** Ensemble photoluminescence lifetime decay for monomer (blue line), unfused dimer (green line), and fused dimer (red line). Time-tagged, time-resolved data for single (**b**) CQD monomer and (**c**) fused dimer. Shown are (from left to right) photoluminescence intensity time trace, second-order photon correlation and lifetime for the single particle respectively (the green and dark gray lifetime curves were generated from data shaded in the same color in the corresponding time traces). The excitation fluence (<$N$>) for monomer and coupled dimer in the represented figures were 0.05 and 0.08, respectively. **d** Possible coupling mechanisms for shortening of lifetime in molecules. **e** Multicarrier configurations. **f** Pump fluence dependency of lifetime for an individual fused dimer. <$N$> values are 0.03 (brown trace), to 0.09 (green trace) and 0.18 (blue trace).

oxide (TOPO, 99%), 1-octanethiol (≥98.5%), pentaerythritol-tetrakis(3-mercapto-propionate) (95%), ammonia aqueous (28.5%), tetrahydrofuran (THF, ≥99.9%), *N*-methylformamide (NMF, 99%), toluene (99.8%), hydrofluoric acid (HF, 48%), polyvinylpyrrolidone (PVP, 10k), ethanol (99%), tetraethyl orthosilicate (TEOS, 98%), hexane (95%) and (3-Mercaptopropyl) trimethoxysilane (MPTMS, 95%) were obtained from Sigma Aldrich. Trioctylphosphine (TOP, 97%) was purchased from Strem Chemicals. Octadecylphosphonic acid (ODPA, >99% was purchased from PCI synthesis. All the reagents were used as received without further purification.

**CdSe core growth**. Briefly, 60 mg CdO, 280 mg ODPA, and 3 g TOPO were added to a 50 mL flask. The mixture was heated to 150 °C and degassed under vacuum for 1 h. Under argon flow, the reaction mixture was heated to 320 °C to form a colorless clear solution. After adding 1.0 mL TOP to the solution, the temperature was brought up to 350 °C. At this point Se/TOP solution (60 mg Se in 0.5 mL TOP) was swiftly injected into the flask. The reaction was kept at 350 °C for suitable time and then stopped by removal of the heating mantle. The resulting CdSe particles were precipitated with acetone and redispersed in 3 mL hexane for use as a stock solution.

**CdSe/CdS core-shell colloidal QD synthesis**. For the shell growth reaction, a hexane solution containing 200 nmol of CdSe CQDs mixed with ODE (6 mL) and OAm (6 mL). The reaction solution was degassed under vacuum at room temperature for 30 min and at 90 °C for additional 30 min to completely remove the hexane, water, and oxygen inside the reaction solution. Then the reaction solution was heated up to 310 °C under argon flow and magnetic stirring. During the heating, when the temperature reached 240 °C, a desired amount of cadmium (II) oleate (Cd-oleate, diluted in 6 mL ODE) and octanethiol (1.2 equivalent amounts to Cd-oleate, diluted in 6 mL ODE) was injected dropwise into the growth solution at a rate of 3 mL/h using a syringe pump. Upon precursor infusion, 2 mL oleic acid was quickly injected and the solution was further annealed at 310 °C for 30 min. The resulting CdSe/CdS core/shell CQDs were precipitated by adding ethanol, and

then redispersed in hexane. The particles were further purified by additional two precipitation–redispersion cycles and finally suspended in ~2 ml hexane.

**Silica NPs synthesis**. In all, 120 μL MPTMS precursor was mixed with 30 mL ammonia aqueous solution (1%) under strong stirring. After stirring for 1 min, the solution was stored overnight. The SiO$_2$ NPs were collected by centrifugation and dispersed in 10 mL of ethanol. Then the SiO$_2$ solution was mixed with PVP solvent (0.02 g mL$^{-1}$) for 30 min. Finally, the NPs were stored after the cleaning by centrifugation.

**Synthesis of CdSe/CdS@SiO$_2$**. One milliliter of SiO$_2$ NPs (0.0079 g mL$^{-1}$) was mixed with 0.5 nmol CdSe/CdS NPs using vortex for 5 min. Then 5 mL of ethanol was added into the vails to precipitate and remove the unattached NPs. After three washing cycles the final SiO$_2$@CdSe/CdS NPs were redispersed in 5 mL of ethanol.

**Synthesis of SiO$_2$@CdSe/CdS@SiO$_2$**. The CdSe/CdS@SiO$_2$ was dispersed in 5 mL of ethanol. Then 330 μL of ammonia solvent (28.5% wt%) was added into the solution with stirring for 5 min. Thereafter, 50 μL of TEOS was added dropwise into the solution. After stirring for 10 h, the resulting solvent was centrifuged (6000 r.p.m., 5 min) and redispersed in 5 mL of THF.

**Synthesis of dimer-CdSe/CdS@SiO$_2$**. A tetrathiol linker, pentaerythritol-tetrakis (3-mercapto-propionate) (200 μL) was added to the CdSe/CdS@SiO$_2$ solution. Then 0.6 nmol of CdSe/CdS CQDs was added and the solution was stirred in an oil bath at 60 °C overnight. Samples were cleaned by centrifugation (6000 r.p.m., 5 min) and redispersed with 10 mL of THF for storage.

**Release of dimer-CdSe/CdS**. One milliliter of dimer-CdSe/CdS@SiO$_2$ CQDs was centrifuged (5000 r.p.m., 5 min), and later mixed with 2 mL of mixed HF/NMF solvent (10%) under stirring for 10 h. Upon etching, the color of the samples

changed to light yellow, which indicates on the removal of the $SiO_2$. Thereafter, the samples were precipitated by centrifugation (6000 r.p.m., 10 min) and washed twice. Finally, the samples were redispersed in 2 mL of ethanol.

**Synthesis of fused dimer-CdSe/CdS CQDs.** Dimer-CdSe/CdS CQDs (in 2 mL of ethanol) were mixed with 2 mL of ODE, 100 μL of Cd-oleate (0.2 M), and 50 μL of OAm. The reaction solution was degassed under vacuum at room temperature for 30 min and again at 90 °C for additional 30 min. Later, the reaction mixture was heated to 180 °C for 20 h under argon flow. The resulting fused particles were precipitated with ethanol and redispersed in 2 mL toluene for use as a stock solution.

**Optical and structural characterization.** Absorption spectra were measured using a Jasco V-570 UV-Vis-NIR spectrophotometer. Fluorescence spectra and ensemble lifetimes were measured with a fluorescence spectrophotometer (Edinburgh instruments, FL920). Transmission electron microscopy (TEM) was performed using a Tecnai $G^2$ Spirit Twin T12 microscope (Thermo Fisher Scientific) operated at 120 kV. High-resolution TEM (HRTEM) measurements were done using a Tecnai F20 $G^2$ microscope (Thermo Fisher Scientific) with an accelerating voltage of 200 kV. High-resolution STEM imaging and elemental mapping was done with Themis Z aberration-corrected STEM (Thermo Fisher Scientific) operated at 300 kV and equipped with HAADF detector for STEM and Super-X EDS detector for high collection efficiency elemental analysis. CQDs atomic structure model were built by the VESTA software. Scanning electron microscopy imaging (SEM) was done with HR SEM Sirion (Thermo Fisher Scientific) operated at 5 kV.

**Single particle optical measurements.** Single particle measurements were performed with an inverted microscope (Nikon Eclipse-Ti) in epi-luminescence configuration. Dilute solution of QDs in 2 wt% poly(methyl-methacrylate) were spin coated on glass coverslips (no.1, precleaned and thermally annealed) leading to minimum separation of 4–5 μm between the dots as confirmed by wide field fluorescence microscopy. The excitation light from a pulsed diode laser (EPL375; Edinburgh Instruments) at a repetition rate of 5 MHz was focused onto single particles through an oil immersion objective (×100; 1.4 NA), which was also used for collecting the emission. The emission light was passed through a dichroic mirror (T387lp) and additional longpass filter (580LP) before focusing onto two Avalanche Photodiodes (Perkin-Elmer; SPCM-AQRH-14) in a Hanbury-Brown-Twiss geometry. Photon statistics of the signal from the detector were performed using multichannel Time Tagger 20 (Swabian Instruments). Time traces, fluorescence lifetime, and second-order photon correlation were extracted from the time-tagged time-resolved data by using home written MATLAB code.

## Data availability
The authors declare that all the important data to support the findings in this paper are available within the main text or in the supplementary information. Extra data are available from the corresponding author upon reasonable request.

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

## Acknowledgements
The research leading to these results has received financial support from the European Research Council (ERC) under the European Union's Horizon 2020 research and

innovation programme (grant agreement No [741767]). J.C and S.K acknowledge the support from the Planning and Budgeting Committee of the higher board of education in Israel through a fellowship. U.B. thanks the Alfred & Erica Larisch memorial chair. Y.E.P. acknowledges support by the Ministry of Science and Technology & the National Foundation for Applied and Engineering Sciences, Israel.

## Author contributions

U.B. oversaw and managed the research. J.B.C., Y.E.P., S.K., M.O. and U.B. designed the experiments. J.B.C. performed the synthesis and materials characterization. S.K and Y.E.P carried out single molecules experiments. Y.E.P. and D.S performed the calculation. S.R. and I.P. assisted with HR-STEM measurements and structural analysis. J.B.C., S.K., Y.E.P., N.W. and U.B. wrote the manuscript with help from the other authors.

## Competing interests

The authors declare no competing interests.
