## [Peer Review File · Nature Communications]

Reviewers' Comments:

Reviewer #1:

Remarks to the Author:

The working manuscript composed by Cui, Panfil and Koley describe the dimerization of CdSe/CdS core shell nanocrystals. Through a nice process of linking and fusing, the work shows the ability to create QD dimers.

I am generally really positive for this work to be published. It would be easily acceptable in a workhorse journal such as Nano Letters or Advanced Materials. Nature Communications should be similarly regarded but should in general have appeal to a larger audience with less specified backgrounds.

Thus I simply recommend the authors to more clearly streamline the reporting of the work.

First the broader appeal:

I like the broad introduction and the analogy to coupling of atoms, but I find the analogy stronger with the coupling of molecules rather than single atoms. What is shown here is much more similar to fusing of molecules like benzene to naphthalene than that of H to H₂ or C to C₂.

Really there isn't a clear roadmap of how one would make a large molecule from single QDs like one would do with atoms to a molecule. So both in the abstract and in the introduction, I recommend to consider reworking the analogy to something along the lines of dimerization of molecules and molecular chemistry rather than atoms to molecules.

Also along the lines of the broader appeal, I would like a stronger visualization of how this would directly impact any of the technologies that are listed. Why specifically is dimerization of QDs important for LEDs, photovoltaics, displays or sensing? I see how the new toolset could be interesting, but I don't see the big picture of how the authors envision this work making any impact on those technologies. Perhaps painting that picture better, rather than just namedropping the technologies would suffice.

Secondly on the reporting:

While I read the paper and find no objectionable data presented, the story isn't written as easily digestible as it could be. For instance in the 3rd paragraph, I hoped I was going to be told what specifically is accomplished here and how to understand the new findings. However it seems to wander between a generalized concept and what has been done by others.

I recommend that this work will be better understood if the findings were clearly laid out. which lattice planes attach, how much the coupling decreases the confinement etc...

This work is likely only a start for this field. One would also want to systematically increase/decrease the shell thickness and look at coupling of the cores with very finely tuned shell thicknesses.

Increase/decrease the core size. combine different sized QDs for heterodimers. Expand from dimers to controllable trimers and larger polymers (similar to ref 26), etc.

In this work the field isn't specifically laid out very well as there is some description of coupling in MBE systems, but many of the references of very similar work with PbSe for example like ref 23-26 aren't described in detail to put the work in perspective. For instance the dimers of PbSe from ref 25 show a clear splitting of the first exciton peak from coupling. The honeycomb lattices from ref 26 show dirac like properties.

Hopefully these comments are helpful and I think this is really nice work, but the conclusions and the reporting of the story could be greatly improved.

Reviewer #2:

Remarks to the Author:

The paper is of interest. The experiment is a tour de force, with multiple steps of synthesis in order to obtain a fraction of CdSe/CdS fused QD dimers that can then be optically probed. A key idea to make the dimers was taken from ref. 20, but there is also a lot of creative nanochemistry involved, such as extracting the QD dimers from Silica using HF, and fine-tuning the proper fusing procedure. The characterization using TEM is extensive at all stages of the fabrication. Any group pursuing the same goal will have to have those high standards.

At the end, the result is somewhat expected, and specific features of a quantum mechanical object may be lacking. Indeed the optical spectra, the lifetimes, the g_2 , are similar in the fused vs attached dimers, even though the QM coupling should be vastly different. I am also questioning whether the g_2 going from ~ 0.1 to ~ 0.6 is as simple an effect of having two (or more) independent chromophores in the same confocal spot.

Overall, a tour de force in synthesis. I feel that it may not be as profound as one could expect, since, in my opinion, the authors made impressive QD dimers, but may not have really made QD "molecules". I am not calibrated with Nat. Comm. but I would recommend this paper to ACS nano for example, as is.

Reviewer #3:

Remarks to the Author:

This manuscript uses an inventive multi-step synthetic approach to generate quantum dot dimers in solution made from CdSe-CdS core-shell quantum dots. It contains detailed structural characterization of the dimers, including very interesting results of the relative crystallographic orientations of the quantum dots within the dimers. Steady-state and time-resolved optical measurements, on ensembles and single particles, were performed with exhaustive controls. Adequate experimental details are provided in the SI. This careful and extensive characterization, well beyond what previous work has applied to quantum dot dimers, will be much valued by researchers in the field of colloidal quantum dots.

The data do show that both dimerization (with linker) and fusion of the dimer each modify the photophysics of the quantum dots; however, the authors discuss their work by analogy to the chemistry of homonuclear molecules and claim that the quantum dots in the dimers exhibit wavefunction hybridization, a claim that is specifically highlighted in the title "Colloidal Quantum Dot Molecules Manifesting Quantum Coupling at Room Temperature." Such terminology connotes a much stronger interaction between two quantum systems than the data show. For example, ref. 25, which is cited only as an example of oriented attachment, reported wavefunction-hybridized PbSe quantum dot dimers and clear evidence of a new excitonic transition in the absorption spectrum. Reaching such hybridization is of course much more challenging in the cadmium system; however, without such a clear signature of hybridization, the emphasis on "quantum coupling" seems forced. The fundamental challenge of moving the quantum dots close enough together for hybridization has not been overcome in this work, but the data provided, in particular the blinking statistics and correlation measurements, are a notable advance for the field.

In analyzing the evidence used to support the idea of wavefunction hybridization, please note the following observations:

1. Figure 1i-j: the redshift of the excitonic absorption and PL peaks. It is well known that changes in the dielectric environment and strain can cause such shifts, and that the magnitude of these effects commonly increases for smaller nanocrystals. See, for example, Chen et al. Surface Functionalization-Dependent Optical Properties of II-VI Semiconductor Nanocrystals, JACS 2011, 133, 17504-17512, doi: 10.1021/ja208337r) and Smith et al. Tuning the Optical and Electronic Properties of Colloidal Nanocrystals by Lattice Strain. Nature Nanotech. 2009, 4, 56-63, doi: 10.1038/nnano.2008.360) This data alone cannot prove that the quantum dots are coupled.
2. Figure 5a: decrease of the ensemble lifetime from monomer to linked dimer to fused dimer. As the authors note, both resonant energy transfer and tunneling might explain the increased decay rates observed with dimerization; however, neither of these mechanisms is "wavefunction hybridization" as claimed in the abstract/introduction. Furthermore, the fused bridge allows only marginally more coupling than the organic linker, rather than the dramatic improvement alluded to earlier on p. 3: "the small distances between the QDs resulting in quantum mechanical coupling an order of magnitude larger than prior systems that is well resolved even at room temperature." Lastly, given resonant energy transfer's sensitivity to dielectric environment and dipole-dipole orientation, it seems premature to dismiss its role in the increased decay rates for the smallest quantum dots.
3. Figures 5b-c: change in blinking dynamics, and increase in biexciton fluorescence yield in dimers relative to monomers. Since the dimer contains two quantum emitters, even if they were totally uncoupled, would you not expect some increase in the $g_2(0)$ value simply because you no longer have a single emitter?

Additional points to address:

1. The generality of this method is claimed based on the different sizes of quantum dots used. As only one chemical system (CdSe-CdS) was investigated, the claim of generality is unwarranted.
2. In the text, the energy separation between the two electron states without Coulombic interaction is given (23 meV) but not the separation between the two electron states with Coulombic interaction. What is this value? Perhaps it could tell you if you expect to see any other excitonic transitions from the dimer.
3. Please include the total number of dimers measured in the caption to Figure 3k. It might even be better to change the y axis to "Number of dimers" instead of %.
4. In the Se EDS map in figure 2c, it looks like the CdS shell might be slightly alloyed with Se (or if an artifact, please account for it), and there appears to be a small increase in the lifetime decays between the monomers that were treated in the fusion reaction (Fig. S17). Please clarify if the monomers measured in figure 1c were left over at the end of the fusion process or were the initial particles. If they were the initial particles, please include an absorption/PL plot for the monomers that went through the full fusion reaction - just to make sure that S-Se alloying did not cause any redshift.
5. Please provide references for the ligand coverage of the (10-10) and (10-1-1) facets of CdS. That information does not seem to be included in reference 28 of the previous sentence.
6. The color label for all of the Cd atoms in the structural models (in the text and SI) is given as brown when the atoms are actually purple.

To summarize, the quality of the experiments in this work is excellent and the concept of quantum dot "molecules" somewhat novel – the results are simply oversold in the current form of the manuscript. Given the ambiguous results, a discussion of what size and separation CdSe/CdS core-shell nanocrystals might be expected to show significant hybridization could be appropriate. Perhaps theory (similar to that in ref. 25) could assist you in this estimation.

Response to the reviewers

Major Modifications at a glance:

1. We thoroughly modified the introduction section to elaborate the analogy of the CQDs with artificial atoms as suggested by the reviewers. In the modified manuscript we have provided sub-headings to each observation for better comprehension. We compared the observations with what has been done before in an elaborate manner. The probable applications are described in a precise way in the revised manuscript.
2. The optical signatures for coupling: The wavefunction hybridization is directly reflected into the band edge transition. Tuning core and shell dimensions leads to tuned electronic coupling strongly supporting these claims. We have added a much more detailed discussion, comparing the calculated and observed red-shift as a function of extent of hybridization. In the same section we have explained how other factors could affect the red shift in CQDs and how the possibilities can be ruled out in our case with proper control experiments. Additional attributes of the coupling signified by broadening of the transitions and the changed absorption in high energies only upon fusion are further emphasized.
3. A detailed description about the PbSe coupled system work and the difference from our system and approach has been included in the revised manuscript.
4. The photophysics and $g_2(0)$ value in the single particle studies. We understand the concerns of the reviewers regarding the correlation between photophysical data and coupling in dimers. The photophysics got altered significantly in the coupled system but this is more than only hybridization. The other photophysical processes such as FRET and charge transfer affects even without wavefunction hybridization, and is present in unfused dimers. The change between the unfused and the fused dimer in term of photophysics has been explained along with the observed $g_2(0)$ value in both systems in the revised manuscript. We emphasize that the shift in the band edge transitions (absorption and emission peaks) are a direct consequence of the extent of coupling in the CQD molecules, while the photophysical data are the results of additional photo-induced processes along with the hybridization.
5. A conclusion section with the main findings along with the future outlook were added as well.

The detailed responses for every concerns are described as following.

REVIEWER 1

The working manuscript composed by Cui, Panfil and Koley describe the dimerization of CdSe/CdS core shell nanocrystals. Through a nice process of linking and fusing, the work shows the ability to create QD dimers. I am generally really positive for this work to be published. It would be easily acceptable in a workhorse journal such as Nano Letters or Advanced Materials. Nature Communications should be similarly regarded but should in general have appeal to a larger audience with less specified backgrounds. Thus I simply recommend the authors to more clearly streamline the reporting of the work.

Response: We thank the reviewer for the positive assessment of the manuscript acknowledging the innovative aspects of the work. In this manuscript we established a general method for the fabrication of QD dimers. Furthermore we presented rigorous structural characterization and examined experimentally and theoretically the attributes of the quantum mechanical coupling in the dimer CQD molecules. This method can be generally utilized to prepare a library of dimers from the variety of semiconductor core-shell nanocrystal building blocks. Addressing the comment regarding the general appeal of the manuscript - we have modified the introduction section as detailed below.

Specific comments:

1. *I like the broad introduction and the analogy to coupling of atoms, but I find the analogy stronger with the coupling of molecules rather than single atoms. What is shown here is much more similar to fusing of molecules like benzene to naphthalene than that of H to H₂ or C to C₂. Really there isn't a clear roadmap of how one would make a large molecule from single QDs like one would do with atoms to a molecule. So both in the abstract and in the introduction, I recommend to consider reworking the analogy to something along the lines of dimerization of molecules and molecular chemistry rather than atoms to molecules.*

Response: Indeed the quantum dots, as artificial atom-like building blocks, differ from the real atoms in many aspects. First of all, the QDs are composed of hundreds to thousands of atoms, and correspondingly are much larger than real atoms (few nm scale with respect to 0.1nm scale for real atoms). Secondly the potential governing electron orbital energies in real atoms is the Coulomb interactions, whereas in QDs the confinement potential is dominant. Nonetheless, The concept of the elegant analogy of semiconductor quantum dots to artificial atoms is well established in the literature. (e.g. R. C. Ashoori, *Nature* 1996 **379**, 413–419), in-line with the beauty of the universality of such scientific concepts.

In particular, spherical colloidal QDs possess atomic-like electronic wavefunctions. The spherical harmonics thus also describe the CQD wavefunctions leading to states with *s* and *p* character related to the spherical symmetry. This is manifested also experimentally (Banin, U., Cao, Y., Katz, D. & Millo, O. *Nature* **400**, 542–544 (1999)). In fusion of molecules such as benzene to naphthalene, the orbital hybridization is more complex, for example utilizing hybrid molecular orbital (*sp*² in this case). This is indeed more elaborate and stretches the analogy beyond its aim to provide a qualitative understanding of the coupling in CQD molecules.

So - from this well-established artificial atom analogy, the roadmap to nanocrystal chemistry forming CQD molecules evolves naturally. So far the formation of CQD molecules focused mostly on geometrical control, for example, by DNA linkers, but in those structures the quantum coupling is limited by the high potential barriers. In our case, the fusion of the two CQDs provides a controlled barrier and leads to formation of new hybridized quantum states. Yet, the energy scale for the tunneling-splitting in CQD molecules that we observe and calculate is indeed much lower than that for the real atoms in a diatomic molecule. This is expected considering the significant differences between the real and artificial atoms.

Therefore, we maintain that the artificial atom analogy is justified and useful. To better clarify this we revised the introduction making it more accessible as the referee justly pointed out.

Changes Made: Modified and added on page 2 1st paragraph to clarify the atomic characteristics of the QDs: We have rewritten the paragraph which is pasted below with the changes highlighted.

"The strongly quantum confined energetic levels of CQDs possess atomic-like character, for example - s and p discrete states, related to their spherical symmetry. This, alongside with the ability to manipulate CQDs into more elaborate structures, naturally led to their consideration as "artificial atoms". Inspired by molecular chemistry, in which functionality of molecules depends on how atoms couple, we apply analogous concepts to enrich CQDs based materials. If one considers CQDs as artificial atom building blocks, how plentiful would be the selection of composition, properties and functionalities of the corresponding artificial molecules? Herein we introduce the utilization of CQDs as basic elements in nanocrystal chemistry for construction of coupled colloidal nanocrystals molecules focusing on homodimer quantum dots (QDs), in analogy to homonuclear diatomic molecules."

Also in the introduction section on page 3 4th paragraph ,we added discussion regarding the same concern.

"The scale of the hybridization energies and corresponding shifts are significantly lower as compared to diatomic molecules. This is expected considering the much larger dimensions of the CQD building blocks compared to atoms and the different potential energy landscape."

2. *Also along the lines of the broader appeal, I would like a stronger visualization of how this would directly impact any of the technologies that are listed. Why specifically is dimerization of QDs important for LEDs, photovoltaics, displays or sensing? I see how the new toolset could be interesting, but I don't see the big picture of how the authors envision this work making any impact on those technologies. Perhaps painting that picture better, rather than just namedropping the technologies would suffice.*

Response: Dimerization of QDs can lead to many new systems with judicious choice of the monomer counterparts. For example, a long standing interest of a dual colour emission center can be envisioned and realized in a suitable heterodimer CQD molecule. This can have direct impact on tagging and display applications with CQDs. We are considering such application but this requires obviously further significant efforts out of the scope of this first manuscript. We have added suitable text to further clarify these potential applications in the revised manuscript.

Changes Made: Modified and added on page 3 5th paragraph:

"Such CQD molecules bear significant promise for their utilization in numerous applications, including in light-emitting devices, displays, photovoltaics and as sensors. For example, the controlled formation of heterodimers consisting of CQD monomers with varying core sizes is of direct relevance for dual color emission. Similarly, forming a heterodimer with a staggered (type-II) band alignment between the two CQD building blocks is envisioned for electric field sensing."

3. *While I read the paper and find no objectionable data presented, the story isn't written as easily digestible as it could be. For instance in the 3rd paragraph, I hoped I was going to be told what specifically is accomplished here and how to understand the new findings. However it seems to wander between a generalized concept and what has been done by others. I recommend that this work will be better understood if the findings were clearly laid out. which lattice planes attach, how much the coupling decreases the confinement etc...*

Response: In the 3rd paragraph (which is now the 4th paragraph in the revised manuscript), we have added a summary of the main findings. We defer the detailed technical descriptions to the main text in favour of accessibility for the general readership of Nature Comm.

Changes Made: Modified and added on page 3 4th paragraph:

"With such control, using high resolution aberration corrected scanning transmission electron microscopy, we observed and analyzed the orientation relationships including homo-plane-attachment and hetero-plane-attachment in the fusion process. Moreover, the manifestations of quantum coupling were revealed by the broadening and red shift of the band gap transition observed in absorption and photoluminescence, in agreement with the quantum-mechanical calculations for the system."

And also in the preceding section of the same paragraph, the observed results are summarized clearly,

"The coupling also leads to broadening of the excited state transitions of the CQD dimers and the absorption spectrum for the high energy bands is modified as well. The emerging attributes of coupling are also revealed by single nanoparticle spectroscopy studies yielding modified electron-hole recombination rates and single photon statistics in CQD dimers in comparison to monomers."

In the conclusion part we have summarized our observations:

“Conclusions

This study introduced a robust route to the realization of coupled colloidal quantum dot molecules with high purity. The construction strategy for the CQD molecules utilized well controlled CdSe/CdS core/shell nanocrystal artificial atom building blocks constructing homodimers fused via "constrained attachment" of the nanocrystal facets. The synthesized novel CQD dimer structures, emitting in the visible range, exhibited electronic coupling at room temperature signified by the red-shift and broadening of the band-edge transition observed both in absorption and in photoluminescence. This is attributed to the quantum coupling and hybridization of the monomer wavefunctions within the CQD molecules, as supported by quantum mechanical calculations. Broadening and modification of the absorption transitions at higher energies is also related to the coupling effects within the CQD dimer molecules. Furthermore, the photo-physical properties of the CQD molecules at single nanoparticle level exhibit lifetime shortening, fluorescence flickering, and increase in the g_2 value for the photon antibunching compared to monomers. These characteristics indicate the introduction of additional recombination pathways and the rich possibilities for multiexciton configurations in the artificial CQD molecule compared with the monomers, also related to coupling within the system.

Formation of coupled CQD homodimer molecules sets the stage for Nanocrystal Chemistry. Considering the rich selection of size and composition controlled CQDs emphasizes the analogy of these artificial atoms to atoms of the periodic table discovered by Mendeleev 150 years ago. As a future outcome, we foresee the formation of a diverse variety of coupled CQD molecules with prodigious promise for their utilization in numerous optoelectronic, sensing and quantum technologies applications.”

4. *This work is likely only a start for this field. One would also want to systematically increase/decrease the shell thickness and look at coupling of the cores with very finely tuned shell thicknesses. Increase/decrease the core size. combine different sized QDs for heterodimers. Expand from dimers to controllable trimers and larger polymers (similar to ref 26), etc.*

Response: Thank you for sharing our vision for this work serving as a basis for *nanocrystal chemistry*. In the present initial work we have indeed focused on the basic homodimer system varying the CQD monomers core and shell sizes (three representative CQD monomer dimensions are reported). The additional highly interesting cases of heterodimers of different kinds are mentioned as potential expansions of the nanocrystals chemistry field (see response and changes in comment 3). These are out of scope of the present manuscript but will emerge in the future.

5. *In this work the field isn't specifically laid out very well as there is some description of coupling in MBE systems, but many of the references of very similar work with PbSe for example like ref 23-26 aren't described in detail to put the work in perspective. For instance the dimers of PbSe from ref 25 show a clear splitting of the first exciton peak from coupling. The honeycomb lattices from ref 26 show dirac like properties.*

Response: MBE grown coupled quantum dots were reported and are therefore important to describe (*Science*. **278**, 1792–1795 (1997); *Science*. **291**, 451–453 (2001); *Science*. **311**, 636–639 (2006) all referenced in the manuscript). However, MBE grown coupled QDs are typically larger than CQDs and their proximity is also limited leading to small energy scale of coupling suitable only for low temperature operation. Moreover, MBE grown structures are inherently embedded in the host semiconductor matrix while the colloidal quantum dots are free standing in solution allowing their integration into diverse solvents and matrices via chemical modification of their surfaces with obvious relevance for multitude of application scenarios.

In the case of PbSe dimers, indeed hybridization effects were observed and the energy scale reached ~50meV as expected from the lighter electron and hole effective masses. However, we find the case of the CdSe/CdS core-shell CQD homodimer molecule closer to the analogy with the diatomic molecules, due to the presence of a controllable potential barrier between the two cores, dictated by the shell thickness. Please note that in the elegant case of the PbSe dimer, after fusion the product is an elongated structure which resembles the properties of a short nanorod. Additionally, the solution based methodology utilizing oriented attachment to form the PbSe dimers leads to a mixture of monomers, dimers and higher order oligomers reaching a level of ~30% in dimers yield, while the methodology introduced herein is selective and highly controlled for formation of different CQD molecules.

With regards to the highly interesting case of the honeycomb lattice of PbSe (Ref. 26 in prior version), we prefer not to elaborate more as this case may make the story more complex and less accessible to the readers.

Changes Made: Modified and added on page 8:

" PbSe CQD dimers were elegantly prepared via oriented attachment in solution, but even under succinct control of ligands, concentration and reaction conditions, only ~30% dimer fraction, a rod-like colloidal quantum system, could be achieved along with monomer and higher order oligomers. In our template based strategy, high control over dimer formation was achieved by firstly forming a connection by molecular linkers. The molecular linkers however, constrain the initial relative crystal orientations between the two monomers."

REVIEWER 2

The paper is of interest. The experiment is a tour de force, with multiple steps of synthesis in order to obtain a fraction of CdSe/CdS fused QD dimers that can then be optically probed. A key idea to make the dimers was taken from ref. 20, but there is also a lot of creative nanochemistry involved, such as extracting the QD dimers from Silica using HF, and fine-tuning the proper fusing procedure. The characterization using TEM is extensive at all stages of the fabrication. Any group pursuing the same goal will have to have those high standards.

Response: We thank the reviewer for his evaluation of our research work.

Specific comments:

1. *At the end, the result is somewhat expected, and specific features of a quantum mechanical object may be lacking. Indeed the optical spectra, the lifetimes, the g_2 , are similar in the fused vs attached dimers, even though the QM coupling should be vastly different. I am also questioning whether the g_2 going from ~0.1 to ~0.6 is as simple an effect of having two (or more) independent chromophores in the same confocal spot.*

Response: The optical spectra of the unfused and fused dimers differ significantly from each other. The band gap absorption feature and the photoluminescence are broadened and red-shifted upon fusion (Figure 1i). The unfused dimer acts essentially as two monomer units in proximity while the fused dimer is way more than that simple consideration. Correspondingly, upon fusion, higher energy absorption features broaden upon fusion, and the overall absorption spectra of fused and unfused dimers

significantly differ (Figure 1i). If we normalize the absorption at the bulk range (~300nm) it can clearly be seen that the loss in the the inherent absorption feature at band edge appears only after fusion (Supplementary Figure 11 in the revised supplementary). This is directly related to the hybridization of electron wavefunctions, that requires rigorous analysis and beyond the scope of only one publication. A follow up publication from our group on the details of particular excited transitions will provide more insight to this, and is currently under preparation.

Supplementary Figure 11. The normalized absorption spectra of monomers (blue), unfused (green), and fused 1.4/2.1 nm CdSe/CdS CQD molecules when normalized with respect to (a) band-edge peak and (b) bulk transitions (300 nm).

Changes Made: We added the first signature of optical spectra and corresponding red shift upon fusion at the start of the discussion paragraph in page 6. We have also considered other factors that generally can cause in a red shift and ruled out their effect in our system systematically.

" After the fusion step, the resultant CQD dimer leaves an interesting optical signature of a red-shift in the absorption and photoluminescence spectra along with broadening of the band gap and excited state spectral features (Fig. 1 and Supplementary Fig. 11). Generally, there are several factors which can lead to a red-shift: the formation of alloying shell, alteration of the dielectric environment (surface ligands) or interfacial strain. To address these different possibilities, we also studied the spectral properties of the monomers, which underwent the fusion reaction under similar conditions (Supplementary Fig. 9-10), and found them to be identical to the original monomer particles (Supplementary Fig. 12). Hence the possibility of observing a red shift in the band gap transition due to formation of an alloy shell or altered dielectric environment can be ruled out. Furthermore, strain effects and change in the dielectric properties during the fusion procedure can be considered negligible as we did not grow an additional shell, but rather the fused shell material is the same (CdS), and the surface ligands are also the same for the CQDs monomers and dimers. Moreover, no shift was observed after the fusion of the large 1.9/4.0nm CQDs (Fig.1), where dielectric and strain effects, if significant, would be expected to contribute as well. In fact, the red shift in the band gap transitions was found to depend systematically on the alteration of the core size and shell thickness of the monomer counterparts, increasing for small core and shell dimensions (Fig 1j). This is consistent with the difference in the delocalization of the wavefunctions in the various CQDs that lead to different degree of coupling of the corresponding wavefunctions in the CQD molecules.

To this end, we have employed quantum mechanical calculations to visualize the wavefunction hybridization and finally to estimate the expected red-shift in different CQD molecular systems.

Also, in the later part of the same section (page 8) we have added some more discussion.

" An additional signature for the coupling in fused dimers is observed in the absorption spectra at higher energies. Figure 1i shows broadening only upon fusion consistent with coupling forming multiple states in dimers. Furthermore, the spectra normalized at the band gap manifest a significantly stronger relative absorbance in high energies for the fused dimers compared with monomers and unfused dimers (see also supplementary figure 11). This is assigned to the wavefunctions modification in the fused system, which can be considered from a viewpoint of hybridization among the excited states."

Response (continued): Moreover, the photoluminescence lifetime is shortened to a larger extent in the case of fused dimers with small cores and thin shells that manifest stronger coupling. This is shown in Figure 5a for the ensemble of fused versus unfused CQDs, and in the Supporting Information for single particles. This is in contrast with the CQD molecules with larger cores and thicker shells where there is indeed no significant change between the unfused and fused dimers in-line with the weaker coupling also revealed by the quantum mechanical calculations.

With regards to $g(0)$ – this is more intricate. Indeed, even in the unfused dimer, two particles that reside sufficiently close affect the photon statistics (linker size~ 0.3nm). We could consider that two emission centers contributed to the analysis, as we are looking at single dimer, for the sake of a simple explanation. But in the case of the fused dimer the situation is more complicated. The fused dimer the particle should absorb as one unit and the excitation power was kept sufficiently low. So it will be premature to solely assign the $g_2(0)$ value to a two particles effect when the particles are in such strong coupling regime. This was indeed observed when we checked an aggregated spot of monomers where the increase in $g_2(0)$ value was observed, however, it lacked the change in fluorescence lifetime. It is known in the literature that the decrease in the exciton QY can cause a loss in the antibunching (Phys Rev B 90, 035311 (2014)). Even in the presence of additional electron rich coupling agent, such as gold nanoparticles, at close proximity, the $g_2(0)$ value of single QDs was observed to be increased. We still need to perform more experiments in order to verify the actual origin of this $g_2(0)$ value.

Changes made: We have explained more about the change in $g_2(0)$ in page 14 of the main text following all the possibilities.

"An increment in the $g^2(0)$ value was found in the case of dimers in general with the possibilities of either two emission centers in the excitation spot or intrinsic properties of coupled systems. Specifically, in the fused dimers, the particles absorb the light as one unit, and at low excitation regime ($\langle N \rangle \sim 0.1$) the possibility of emission from one of the centers is rational statistically. The full understanding of the $g^2(0)$ value requires more rigorous experimental studies while an interesting multicarrier configuration can be realized in these fused CQD dimers."

2. *Overall, a tour de force in synthesis. I feel that it may not be as profound as one could expect, since, in my opinion, the authors made impressive QD dimers, but may not have really made QD "molecules".*

I am not calibrated with Nat. Comm. but I would recommend this paper to ACS nano for example, as is.

Response: In addition to the rigorous effort in the synthesis of CQD dimers and their structural/chemical characterizations, we correlated each and every particle at different stages (monomer, unfused, fused) with quantum mechanical calculations and detailed optical spectroscopy studies. The synthesis of different sized particles was intended to demonstrate, beyond the generality of the synthetic method, also the possibility for wavefunction engineering that affected the quantum mechanical coupling accordingly. In Fig.1j we have correlated the experimental red-shift in the band edge with the quantum mechanical calculations. The attributes of single QD dimers are revealed by steady state and a time resolved spectroscopic measurements, and several new properties could be observed upon fusion. The wavefunction hybridization and CQD molecule formation is proven in our opinion. The magnitude of hybridization depends strongly on the core size and the barrier thickness. This can be seen in Figure. R1 presenting the dependence of the energy splitting between the bonding and antibonding electron states for CQD dimers upon change in core diameter and barrier thickness. The calculated energy scale is well correlated with the experimental red shift upon fusion. Specifically, this dependence is reflected in the three type of CQDM prepared and studied in the present manuscript.

Figure. R1. A contour plot of the energy splitting between the symmetric and anti-symmetric states for the lowest conduction band levels in the CQDM (the energy values shown in the color scale) as a function of barrier thickness and core diameter. The CdSe-CdS band offset for the calculation was 0.1eV.

REVIEWER 3

This manuscript uses an inventive multi-step synthetic approach to generate quantum dot dimers in solution made from CdSe-CdS core-shell quantum dots. It contains detailed structural characterization of the dimers, including very interesting results of the relative crystallographic orientations of the quantum dots within the dimers. Steady-state and time-resolved optical measurements, on ensembles and single particles, were performed with exhaustive controls. Adequate experimental details are provided in the SI. This careful and extensive characterization, well beyond what previous work has applied to quantum dot dimers, will be much valued by researchers in the field of colloidal quantum dots.

Response: We thank the reviewer for his evaluation.

Specific comments:

1. *The data do show that both dimerization (with linker) and fusion of the dimer each modify the photophysics of the quantum dots; however, the authors discuss their work by analogy to the chemistry of homonuclear molecules and claim that the quantum dots in the dimers exhibit wavefunction hybridization, a claim that is specifically highlighted in the title “Colloidal Quantum Dot Molecules Manifesting Quantum Coupling at Room Temperature.” Such terminology connotes a much stronger interaction between two quantum systems than the data show. For example, ref. 25, which is cited only as an example of oriented attachment, reported wavefunction-hybridized PbSe quantum dot dimers and clear evidence of a new excitonic transition in the absorption spectrum. Reaching such hybridization is of course much more challenging in the cadmium system; however, without such a clear signature of hybridization, the emphasis on “quantum coupling” seems forced. The fundamental challenge of moving the quantum dots close enough together for hybridization has not been overcome in this work, but the data provided, in particular the blinking statistics and correlation measurements, are a notable advance for the field.*

Response: The wavefunction hybridization is indeed expected to be smaller in the case of a CdSe/CdS system as compared to the PbSe QDs (see also response to Ref. 1, comments 5 and changes made to

address the PbSe dimers case). In the context of wavefunction coupling, we note that the core/shell system allows us to tailor the barrier via the shell characteristics (thickness, and in future also its compositions). This type of control is not possible in attaching core QDs.

To add further to coupling effects, we emphasize that a clear difference in the absorption spectrum can be seen when two particles are fused (Supplementary Figure 11.). The unfused dimer acts like simple monomer with twice as value of absorption cross-section and monomer like band edge transition. But upon fusion, the energy of the band edge changes and the higher transitions differ from the monomer and from the unfused dimer as well. We also observed a new excitonic band which is red shifted from the monomer QD.

Changes made:

A detailed discussion about the wavefunction hybridization and the effect on optical spectrum is added in the main manuscript at page number. 6 (Please refer to the response for Reviewer 2 Comment1) and also in the supporting information as follows.

" The features in the absorption spectrum change significantly after the fusion of two CQDs. To further demonstrate the changes, we have normalized the data at the bulk absorption regime (300nm). It can be observed that the band-edge absorption feature of the unfused dimer is retained and similar to the monomers. Upon fusion broadening is seen, along with a red shift and lower absorption. The excited state features are also broadened significantly upon fusion. Upon normalization at the band-edge, the significant relative change in absorption at higher energies upon fusion is emphasized (Supplementary Figure 11a). All these aspects indicate coupling effects upon fusion and formation of the CQD molecules."

Supplementary Figure 11. The normalized absorption spectra of monomers (blue), unfused (green), and fused 1.4/2.1 nm CdSe/CdS CQD molecules when normalized with respect to (a) band-edge peak and (b) bulk transitions (300 nm).

2. *Figure 1i-j: the redshift of the excitonic absorption and PL peaks. It is well known that changes in the dielectric environment and strain can cause such shifts, and that the magnitude of these effects commonly increases for smaller nanocrystals. See, for example, Chen et al. Surface Functionalization-Dependent Optical Properties of II-VI Semiconductor Nanocrystals, JACS 2011, 133, 17504-17512, doi: 10.1021/ja208337r and Smith et al. Tuning the Optical and Electronic Properties of Colloidal Nanocrystals by Lattice Strain. Nature Nanotech. 2009, 4, 56-63, doi: 10.1038/nnano.2008.360) This data alone cannot prove that the quantum dots are coupled.*

Response: Indeed, the dielectric environment and strain can result in the red shift of the absorption and emission spectrum. However, in our system, the surface ligands of the unfused and fused CQD molecules are the same and no shift was observed with the 1.9/4.0 nm CQDs where wavefunction hybridization is negligible. But, the emission and absorption spectrum of the unfused CQDs was similar compared with the monomer unlike the fused dimer. Moreover, during the fusion reaction we did not add any shell precursor to grow another shell which could produce extra strain in the dimer molecule. On the other hand, for the big size CQDs (1.9/4.0 nm) molecules system, the spectrum feature was the same without any change between unfused and fused dimers. This is nicely correlated with the quantum mechanical calculation

where the red-shift is proportional to the coupling of the electron wavefunction. Moreover, the monomer particles recovered from the supernatant of the size selective precipitation process in the fused sample did not exhibit any red shift (please refer to Supplementary Figure 12 in comment 8). Thus the possibility of dielectric environment and the strain on the red shift can be excluded.

Changes made: We have considered all these factors and discussed accordingly in the results section with added references.

" After the fusion step, the resultant CQD dimer leaves an interesting optical signature of a red-shift in the absorption and photoluminescence spectra along with broadening of the band gap and excited state spectral features (Fig. 1 and Supplementary Fig. 11). Generally, there are several factors which can lead to a red-shift: the formation of alloying shell²¹, alteration of the dielectric environment²² (surface ligands) or interfacial strain²³. To address these different possibilities, we also studied the spectral properties of the monomers, which underwent the fusion reaction under similar conditions (Supplementary Fig. 9-10), and found them to be identical to the original monomer particles (Supplementary Fig. 12). Hence the possibility of observing a red shift in the band gap transition due to formation of an alloy shell can be ruled out. The other possibilities of strain effects and change in the dielectric properties during the fusion procedure can be considered negligible as we did not grow an additional shell, but rather the fused shell material is the same (CdS), and the surface ligands are also the same for the CQDs monomers and dimers. Moreover, no shift was observed after the fusion of the large 1.9/4.0nm CQDs (Fig.1), where dielectric and strain effects, if significant, would be expected to contribute as well. "

21. Bae, W. K. *et al.* Controlled Alloying of the Core-Shell Interface in CdSe/CdS Quantum Dots for Suppression of Auger Recombination. *ACS Nano* **7**, 3411–3419 (2013).

22. Chen, O. *et al.* Surface-Functionalization-Dependent Optical Properties of II-VI Semiconductor Nanocrystals. *J. Am. Chem. Soc.* **133**, 17504–17512 (2011).

23. Smith, A. M., Mohs, A. M. & Nie, S. Tuning the optical and electronic properties of colloidal nanocrystals by lattice strain. *Nat. Nanotechnol.* **4**, 56–63 (2009).

3. *Figure 5a: decrease of the ensemble lifetime from monomer to linked dimer to fused dimer. As the authors note, both resonant energy transfer and tunneling might explain the increased decay rates observed with dimerization; however, neither of these mechanisms is “wavefunction hybridization” as claimed in the abstract/introduction. Furthermore, the fused bridge allows only marginally more coupling than the organic linker, rather than the dramatic improvement alluded to earlier on p. 3: “the small distances between the QDs resulting in quantum mechanical coupling an order of magnitude larger than prior systems that is well resolved even at room temperature.” Lastly, given resonant energy transfer’s sensitivity to dielectric environment and dipole-dipole orientation, it seems premature to dismiss its role in the increased decay rates for the smallest quantum dots.*

Response: We observe a notable change in the absorption feature of the QD dimer upon fusion compared to the unfused dimer case where they are only linked to each other in a close proximity. The loss in distinct 1Se and 1P feature in the absorption level upon fusion is indicative of states mixing and that was also reflected in other optical properties.

The radiative recombination rates are related to the interaction among two particles, but we attributed the resonant energy transfer and tunneling as the mechanisms of fluorescence quenching rather than the mechanism of wavefunction hybridization. As per our calculation and experimental observation, fusion of nanocrystal facets is important for wavefunction hybridization and thus, we did not observe any shift in the band edge before fusion, due to the presence of high barrier of the ligands. “*the small distances between the QDs resulting in quantum mechanical coupling an order of magnitude larger than prior MBE grown systems that is well resolved even at room temperature.*” this statement is based on the shift in the band edge energy which is more than 10meV in our case compared to the prior MBE systems where a coupling of a few meV were reported.

The ligands are not a sufficient barrier for the radiative rates in contrast. The linker has a dimension of 0.3 nm and is not expected to resist either resonant energy transfer nor some electron tunneling. This is why we do not observe much change in the decay rate of the fused and unfused dimer as compared to

the monomer. Indeed, the dielectric environment can change the decay rate but the environment of the unfused and fused dimer should remain nearly the same, as a change in the orientation of any of the monomers composing the dimer is unlikely after fusion. This is due to the mode of attachment, which is constrained attachment where the ligand forces the QD to fuse in an orientation as it was in unfused dimer.

Lastly we want to emphasize that the claim of the hybridization was based on the band edge energies and the decay rates were the attributes of the hybridized system.

Changes made: We have clearly discussed the effect of hybridization on the observed red shift as mentioned in the previous comment.

4. *Figures 5b-c: change in blinking dynamics, and increase in biexciton fluorescence yield in dimers relative to monomers. Since the dimer contains two quantum emitters, even if they were totally uncoupled, would you not expect some increase in the $g^2(0)$ value simply because you no longer have a single emitter?*

Response: The photophysics of the unfused and fused dimer looks similar to some extent as they reside sufficiently close to affect the time dependent fluorescence even in the linker connected dimer (linker size~ 0.3nm) About the $g^2(0)$, it is possible that two emission centers contributed to the analysis, for the sake of a simple explanation, but in case of fused dimer the situation is more complicated. It was observed when we check an aggregated spot of monomers where the increase in $g^2(0)$ value was observed without the change in fluorescence lifetime. In case of fused dimer the particle should absorb as one unit and the excitation power was kept sufficiently low. So it will be premature to assign the $g^2(0)$ value to only two particles effect when the particles are in strong coupling regime. It is known in the literature the decrease in the exciton QY can cause a loss in the antibunching (Phys Rev B 90, 035311 (2014)). Even in presence of additional electron rich coupling agent, such as gold nanoparticles, at close proximity, the $g^2(0)$ value of single QDs was observed to be increased. We need more experiment to find the actual origin of this peculiar $g^2(0)$ value.

Changes made: We have explained more about the change in $g^2(0)$ in the page 14 of the main text following all the possibilities.

" An increment in the $g^2(0)$ value was found in the case of dimers in general with the possibilities of either two emission centers in the excitation spot or intrinsic properties of coupled systems. Specifically, in the fused dimers, the particles absorb the light as one unit, and at low excitation regime ($\langle N \rangle \sim 0.1$) the possibility of emission from one of the centers is rational statistically. The full understanding of the $g^2(0)$ value requires more rigorous experimental studies while an interesting multicarrier configuration can be realized in these fused CQD dimers."

5. *The generality of this method is claimed based on the different sizes of quantum dots used. As only one chemical system (CdSe-CdS) was investigated, the claim of generality is unwarranted.*

Response: Our method is based on the linker binding and fusion through oriented attachment. We already established the method with different size of quantum dots based on the different core radius of CdSe (1.2, 1.4, and 1.9 respective) and different shell thickness of CdS (2.1 and 3.9). On the other hand, the oriented attachment procedure is well established for diverse nanocrystals formation such as ZnSe, CdS, PbSe to name a few. With careful tuning of precursor and judicious choice of the fusion conditions, we can foresee high potential for this strategy to work for other colloidal nanocrystal systems as well.

Changes made: The explanation for the generality of our method is added in the main text.

"With careful tuning of precursor and judicious choice of the fusion condition, we can foresee high potential for this method to serve as a general coupling strategy for other colloidal nanocrystal systems."

6. *In the text, the energy separation between the two electron states without Coulombic interaction is given (23 meV) but not the separation between the two electron states with Coulombic interaction.*

What is this value? Perhaps it could tell you if you expect to see any other excitonic transitions from the dimer.

Response: the energy separation of the bonding and anti-bonding state with and without Coulombic interaction is 23 meV, 60 meV, respectively. And the corresponding data was added in manuscript and shown as below:

Changes Made: Modified Fig.1 and added on page 7:

"The Coulombic interaction for the first electron level, localized around the hole, is greater than the second electronic state in the opposite dot, increasing the energy spacing between the bonding and anti-bonding states to 60 meV."

Fig. 1. Coupled CQDs molecule. (a) Scheme for fabrication of coupled CdSe/CdS CQD molecule. (b) The dimer@SiO₂ CQD structure. The dimer 1.9/4.0 nm CQD molecules (c) before, and (d) after the fusion procedure. (e) The 1.4/2.1 nm fused CdSe/CdS CQD molecules. Schematic structures are illustrated. Scale bars (b-e) are 50 nm and insets show higher magnification images. (f) The potential energy landscape and a cross-section of the calculated first electron wave-function without Coulombic interaction Ψ_e (red), with Coulombic interaction Ψ'_e (green) and hole wave-functions Ψ_h (blue) of the coupled CQD molecules. (g) Calculated bonding and antibonding 2- dimensional electron wave-functions without (cross-section of the bonding state is the red curve in f), and (h) with Coulombic interaction (cross-section of the bonding state is the green curve in f). (i) Absorption and fluorescence spectra of monomers (blue), unfused (green), and fused 1.4/2.1 nm CdSe/CdS CQD molecules (red). (j) Calculated (red asterisk) and experimental (blue circles) band gap red shift of monomer-to-respective-homodimer structures for CQD molecules with different core/shell dimensions.

7. Please include the total number of dimers measured in the caption to Figure 3k. It might even be better to change the y axis to "Number of dimers" instead of %.

Response: The total amount of dimers measured in the Fig. 3k was 100. Here, what we pitched on the caption of Fig. 3k was the statistic result for the fusion orientation relationship of the coupled dimer molecules including the homo-attachment and hetero-attachment. So we believe the "percent (%)" was suitable for the y axis.

Changes Made: We modified the figure caption and added a sentence about it on page 10:

"Here, the total amount of dimers for the statistic was 100. Inset shows the CQD model and faces. "

8. In the Se EDS map in figure 2c, it looks like the CdS shell might be slightly alloyed with Se (or if an artifact, please account for it), and there appears to be a small increase in the lifetime decays between the monomers that were treated in the fusion reaction (Fig. S17). Please clarify if the monomers measured in figure 1c were left over at the end of the fusion process or were the initial particles. If they were the initial particles, please include an absorption/PL plot for the monomers that went through the full fusion reaction - just to make sure that S-Se alloying did not cause any redshift.

Response: the alloyed procedure was only possible in high temperature for the shell growth stage. Here, for the fusion step, the reaction condition was at 180 °C which was not suitable for the alloying of the selenium. And for the Se EDS mapping, it may be affected by the background and signal to noise limit of the measurements. Additionally, as shown in the EDS line-scan data in Fig. 2b of the manuscript, the Selenium was confined in the core region without alloyed feature.

Moreover, the absorption and PL spectrum of the monomer recovered from size selective separation step which underwent the fusion procedure is shown below without red-shift. Therefore we conclude that the red-shift of the CQDs molecules was not affected by the S-Se alloying.

Regarding the lifetime figure in Supplementary Figure 20, they are single particle measurements and possess a distribution as shown by the histogram in Supplementary Figure 18. The distribution of the lifetimes from both types of monomers are similar and it would be misleading to infer from a single lifetime. More sample curves are added to the same figure in the revised manuscript.

Changes Made: The fixed figure was add in Supplementary Figure S12 with added discussion in the main text (Reviewer 3 Comment 2).

" Generally, there are several factors which can lead to a red-shift: the formation of alloying shell, alteration of the dielectric environment (surface ligands) or interfacial strain. To address these different possibilities, we also studied the spectral properties of the monomers, which underwent the fusion reaction under similar conditions (Supplementary Fig. 9-10), and found them to be identical to the original monomer particles (Supplementary Fig. 12). Hence the possibility of observing a red shift in the band gap transition due to formation of an alloy shell can be ruled out. The other possibilities of strain effects and change in the dielectric properties during the fusion procedure can be considered negligible as we did not grow an additional shell, but rather the fused shell material is the same (CdS), and the surface ligands are also the same for the CQDs monomers and dimers. Moreover, no shift was observed after the fusion of the large 1.9/4.0nm CQDs (Fig.1), where dielectric and strain effects, if significant, would be expected to contribute as well."

And also in the supporting information, the discussed results are summarized clearly,

"The CQDs are exposed to different conditions during the synthetic procedure such as binding, etching, fusion etc. An important control is to identify whether the inherent properties of the CQDs change during these processes, especially during etching and fusion. Hence, we studied the absorption and photoluminescence spectra (Supplementary Figure 12) of monomers that although did not form a dimer, were still treated with the complete synthetic procedures of the fusion protocol. These monomers were separated by size selective precipitation (Supplementary Figure 10 b) and measured accordingly. The identical spectral features for the monomers, that underwent the fusion procedure, and the original ones

unequivocally confirms no change in the excitonic behavior of the CQDs. This rules out the possibilities of any shift due to alloying between core/shell regions."

Supplementary Figure 12. The ensemble absorption and PL intensity of monomer (black), and monomer in the fusion procedure (red) CdSe-CdS CQDs with core/shell size of 1.2/2.1 nm.

9. Please provide references for the ligand coverage of the (10-10) and (10-1-1) facets of CdS. That information does not seem to be included in reference 28 of the previous sentence.

Response: The literature for the ligands coverage of CdSe@CdS was provided: *J. Am. Chem. Soc.* 2012, 134, 18585–18590; *J. Am. Chem. Soc.* 2000, 122, 12700–12706. The corresponding paper was cited and fixed in the manuscript.

Changes made: Modified and added on page 11 1st paragraph:

"Both (10 $\bar{1}$ 0) and (10 $\bar{1}$ 1) facets plentiful but better passivated³²⁻³³."

32. Ithurria, S. & Talapin, D. V. Colloidal Atomic Layer Deposition (c-ALD) using Self-Limiting Reactions at Nanocrystal Surface Coupled to Phase Transfer between Polar and Nonpolar Media. *J. Am. Chem. Soc.* **134**, 18585–18590 (2012).

33. Manna, L., Scher, E. C. & Alivisatos, A. P. Synthesis of soluble and processable rod-, arrow-, teardrop-, and tetrapod-shaped CdSe nanocrystals. *J. Am. Chem. Soc.* **122**, 12700–12706 (2000).

10. The color label for all of the Cd atoms in the structural models (in the text and SI) is given as brown when the atoms are actually purple.

Response: The colour label for Cadmium, Selenium, and Sulfur was fixed in the SI and the changed figure was shown as below:

Changes Made: The fixed figure was add in Supplementary Figure S3:

Supplementary Figure 3. Structural characterization of the coupled NCs. Raw (a) and Fourier filtered (b) HAADF-STEM images of 1.9/4.0 nm CdSe/CdS CQD monomer viewed under $[\bar{1}2\bar{1}0]$ zone axis (ZA). Inset in (a) is a cartoon model built with VESTA software with bounding faces indexed based on STEM data. Magnified images of edge (shell) (c) and central (core) parts (d) of the CQD shown in (b). Sulfur, Selenium and Cadmium atoms are marked in blue, green and purple, respectively. Coherent growth of the shell lattice is identified. (e) and (f) are FFT and atomic structure model of (a), respectively. HAADF-STEM image of CdSe/CdS CQD under ZA $[0001]$ (g) and atomic structure reconstruction imaging calculated for the same orientation (h). (i) High resolution HAADF-STEM image and atomic structure model (j) of CdSe/CdS CQD viewed under ZA $[\bar{1}2\bar{1}0]$. The core regions are marked with pink circles in (g) and (i). FFT patterns are inserted in (g), and (i). SAED (k) and XRD pattern acquired at large ensembles of CdSe/CdS CQDs (blue curve – experimental XRD data, red bars - theoretical positions for diffraction peaks of *hcp* CdS (JCPDF 04-001-6853), black curve - integrated intensity of SAED (k)).

Reviewers' Comments:

Reviewer #3:

Remarks to the Author:

Thank you to the authors for addressing the reviewers' comments thoroughly. I have no objection to the manuscript's publication in its revised form.